# Sizing nanomaterials in bio-fluids by cFRAP enables protein aggregation measurements and diagnosis of bio-barrier permeability

Ranhua Xiong[1,2], Roosmarijn E. Vandenbroucke[3,4], Katleen Broos[5], Toon Brans[1,2], Elien Van Wonterghem[3,4], Claude Libert[3,4], Jo Demeester[1], Stefaan C. De Smedt[1,*] & Kevin Braeckmans[1,2,*]

Sizing nanomaterials in complex biological fluids, such as blood, remains a great challenge in spite of its importance for a wide range of biomedical applications. In drug delivery, for instance, it is essential that aggregation of protein-based drugs is avoided as it may alter their efficacy or elicit immune responses. Similarly it is of interest to determine which size of molecules can pass through biological barriers *in vivo* to diagnose pathologies, such as sepsis. Here, we report on continuous fluorescence recovery after photobleaching (cFRAP) as a analytical method enabling size distribution measurements of nanomaterials (1–100 nm) in undiluted biological fluids. We demonstrate that cFRAP allows to measure protein aggregation in human serum and to determine the permeability of intestinal and vascular barriers *in vivo*. cFRAP is a new analytical technique that paves the way towards exciting new applications that benefit from nanomaterial sizing in bio-fluids.

[1] Department of Pharmaceutics, Laboratory of General Biochemistry and Physical Pharmacy, Ghent University, Ghent 9000, Belgium. [2] Centre for Nano- and Biophotonics, Ghent University, Ghent 9000, Belgium. [3] Inflammation Research Center, VIB, Ghent 9052, Belgium. [4] Department of Biomedical Molecular Biology, Ghent University, Ghent 9000, Belgium. [5] Anabiotec nv, Evergem 9940, Belgium. * These authors contributed equally to this work. Correspondence and requests for materials should be addressed to K.B. (email: Kevin.Braeckmans@UGent.be).

Measuring the size of nanosized materials in complex biological fluids, such as blood or cerebrospinal fluid, is of great importance in a wide range of applications in the life sciences. In drug delivery, for instance, the effective size of nanomaterials in bio-fluids is important because it directly influences the biodistribution in the body[1–4]. Indeed, even though nanomedicine formulations may be stable under normal storage conditions, they may very well aggregate after administration into a biological fluid such as blood[5,6]. Similarly, there is a growing appreciation that the colloidal stability of therapeutic proteins needs to be tested in blood as protein aggregation after intravenous administration will alter their functionality and may induce immunogenic responses[7,8]. Yet, methods to investigate submicron protein aggregates in serum are virtually non-existent[9,10]. Being able to size nanomaterials in bio-fluids is of interest to medical diagnosis as well, for instance to determine intestinal or vascular barrier permeability which is related to several pathologies, such as sepsis, liver disease, inflammatory bowel disease and neurodegenerative diseases[11–13]. Barrier permeability can be assessed by administering inert size probes, for example, orally or intravenously, followed by quantification of the size and amount of probes that have leaked through the barrier.

Despite its relevance, measuring the size of molecules and nanomaterials in complex biological fluids remains a major challenge. A few years ago our group demonstrated that nanoparticles can be sized in undiluted biological fluids by fluorescence single particle tracking microscopy[14–17]. However, as it is based on imaging the Brownian motion of individual, fluorescently labelled nanomaterials, it is mostly suited for nanoparticles with a size above $\sim 0.1\,\mu m$. Therefore, a technique for measuring size distributions of nanomaterials in bio-fluids in the 1–100 nm range is still very much needed.

Here we report on the use of fluorescence recovery after photobleaching (FRAP) to measure size distributions of nanomaterials in biological fluids. In a FRAP experiment, the sample is placed on a confocal laser scanning microscope and the fluorescently labelled molecules or nanoparticles are photobleached in a micron sized area by a powerful excitation pulse. The fluorescence inside the bleach area will subsequently recover at a rate that is proportional to the diffusional rate of the fluorescent species. Until now, FRAP data have mostly been analysed and interpreted in terms of a single average diffusion coefficient. Verkman and Periasamy[18] were the first to develop a FRAP model for the measurement of continuous distributions of diffusion coefficients. The method was based on measuring the fluorescence intensity as a function of time in a spot bleached by a stationary focused laser beam, as was common at that time. Consequently, since only time information was taken into account, the resolution to discriminate species with a different diffusion coefficient was rather limited (factor of 8). A similar approach was recently reported based on multi-photon spot beaching experiments for determining the size of macromolecular complexes in cells[19]. In the meantime, Hauser et al.[20] showed that the resolution to discriminate two diffusing components could be substantially enhanced (factor of 3) by including spatial information into FRAP analysis. Building forth on these concepts, here we propose an improved FRAP methodology that enables the measurement of continuous distributions of diffusion coefficients (cFRAP), which can be easily converted to equivalent size distributions. A rectangle is photobleached and the full tempo-spatial information available in the confocal recovery images is exploited using a dedicated theoretical recovery model to extract a continuous distribution of diffusion coefficients. The method is very flexible in that the rectangle can have any size, which conveniently allows to optimize the recovery time for a given diffusion coefficient so as to optimally match the sampling rate of the microscope used.

Following detailed validation of the cFRAP-sizing approach, we demonstrate its strength and versatility in a number of challenging sizing applications. First we demonstrate that cFRAP-sizing enables accurate determination of protein aggregation in undiluted blood serum. Next, in combination with the administration of a broad range of inert size probes, we show that cFRAP-sizing allows to characterize in great detail the intestinal and vascular permeability in mice. Importantly, since a single measurement is sufficient to determine the full-size distribution of probes that have leaked through the barrier, we find that the number of animals needed to assess the barrier permeability is reduced up to five times compared with classic approaches where probes of different size are administered and analysed separately.

## Results

**Validation of cFRAP-sizing.** The cFRAP method that we propose is based on the photobleaching of a rectangular area (Fig. 1 and Supplementary Fig. 1). A time-lapse confocal image series is recorded after photobleaching to capture the fluorescence recovery due to diffusion. A closed-form analytical model describing continuous diffusion in such a bleached rectangle (see the 'Methods' section) is fitted in a least-squares sense to this three-dimensional (3D) data set (two-dimensional (2D) spatial $+1$-D temporal) under condition of maximum entropy. The maximum entropy criterion ensures that a continuous distribution of diffusion coefficients is obtained with no more features than statistically warranted by the data[14,21]. In its most straightforward implementation, the cFRAP model is fitted to all individual pixels in the recovery images. However, by averaging the spatial information in rectangular rings, we found that the same precision was obtained with a reduction of the calculation time by about three orders of magnitude (cfr. Supplementary Fig. 2 and Supplementary Note 1). Following optimization of the relevant experimental parameters (Supplementary Figs 3 and 4, and Supplementary Notes 2 and 3), we found from simulations that cFRAP can distinguish two subpopulations if their diffusion coefficient differs by as small as a factor 3 (Supplementary Fig. 5a and Supplementary Note 4). In comparison, in classic FRAP where only the average fluorescence in the bleach area is considered as a function of time, the diffusion coefficient of both subpopulations should differ by at least a factor of 8 (Supplementary Fig. 5b and Supplementary Note 4). Next, we confirmed through simulations that cFRAP can correctly analyse polydisperse systems with a continuous broad range of diffusion coefficients (Supplementary Fig. 6 and Supplementary Note 5). Furthermore, we investigated what is the minimal signal to noise ratio that is needed in the recovery images to perform meaningful cFRAP analysis. Based on simulated data as well as experimental data on a dilution series of fluorescein isothiocyanate (FITC)-dextran of 40 kDa (FD40), we found that cFRAP analysis can be performed down to signal to noise ratio $= 2.4$ (Supplementary Fig. 7). For FD40 this corresponded to a lower concentration limit of $4\,\mu g\,ml^{-1}$ (100 nM) on the microscope used in this study.

The performance of cFRAP-sizing was compared experimentally to dynamic light scattering (DLS) as a standard technique for measuring the size distribution of nanomaterial dispersions. Solutions of dextrans of various molecular weights were prepared and their size distribution was measured by DLS. FITC-labelled dextrans of similar molecular weights were used for sizing by cFRAP. In all cases the cFRAP size distributions corresponded very well with the ones obtained by DLS (Supplementary Fig. 8). As could be expected, thanks to including spatial information in

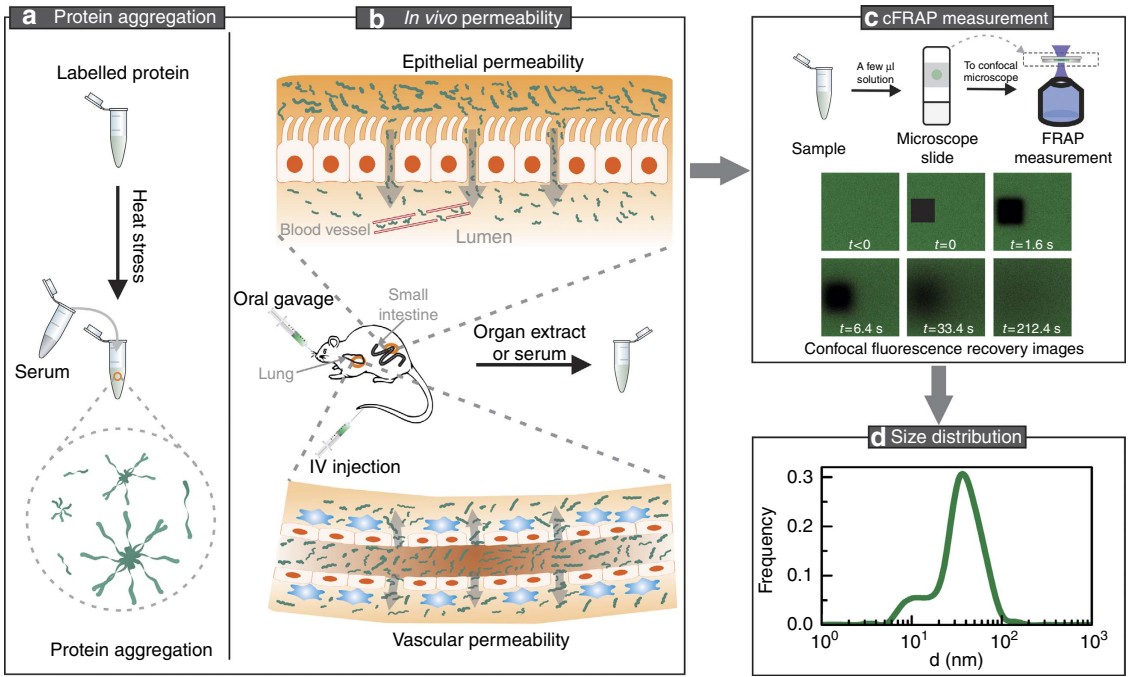

**Figure 1 | Schematic overview of cFRAP-sizing experiments.** (**a**) Measuring protein aggregates in serum and (**b**) measuring the permeability of the small intestines and vasculature of mice following oral gavage or intravenous (IV) injection of fluorescent probes. (**c**) Only a few microliters of sample are required for cFRAP experiments on a standard confocal microscope (**d**) for retrieving the size distribution.

the cFRAP model, the polydispersity index (PDI) of the distributions was significantly less compared with the apparent PDI measured by DLS which only takes time information into account and, therefore, has more limited precision (Supplementary Fig. 9).

Next, we wanted to prove experimentally that cFRAP is very well capable of analysing broad size distributions of nanomaterials. Therefore, as a final validation step, we prepared mixtures of FITC-dextran (FD) with a gradually increasing range of MW to see if cFRAP can measure the full-size distribution correctly. As the results in Fig. 2 show, cFRAP can accurately retrieve the expected size distributions from $\sim 2$ to $\sim 80$ nm, in line with the aim to develop a method for nanomaterial sizing in this range.

**Characterization of protein aggregation in biological fluids.** As a first application, we used cFRAP-sizing to analyse protein aggregates in the sub 0.1 μm range in full serum (Fig. 1). This is of current interest since protein aggregation has emerged as a key issue underlying multiple deleterious effects in the use of protein therapeutics, including loss of efficacy, altered pharmacokinetics, reduced stability and shelf life, and induction of unwanted immunogenicity[7,8]. Fluorescently labelled bovine serum albumin (BSA) was used as a model protein, which could be analysed by cFRAP down to a concentration of $4\,\mu g\,ml^{-1}$ (60 nM; Supplementary Fig. 10). Protein aggregates were prepared by applying heat stress to the BSA monomers in a buffer solution. The unstressed and heat-stressed samples were first characterized in buffer solution by two standard techniques: DLS and size exclusion chromatography (SEC). For the heat-stressed sample, SEC showed both a monomer peak and the presence of aggregates (Fig. 3a). However, the SEC signal corresponding to aggregates was not very well resolved, so that the extent of aggregation was difficult to assess. On the other hand, DLS does show the size range of aggregates in the heat-stressed sample but

failed to detect the monomers (Fig. 3b). Interestingly, cFRAP-sizing could discriminate both monomers and aggregates in a single measurement instead (Fig. 3c), with the size range of aggregates in excellent agreement with the DLS size distribution. This nicely demonstrates again the superior resolving power of cFRAP. Next, we addressed the main question if cFRAP is able to size protein aggregates directly in serum, which is difficult, if not impossible, to do by DLS or SEC. cFRAP was used to analyse samples prepared of BSA monomers and BSA aggregates in 90% serum. Comparison with the size distributions in buffer solution shows that both monomers and aggregates could be correctly sized in serum by cFRAP (Fig. 3d). We conclude that cFRAP-sizing is very well capable of accurately quantifying protein aggregates in a complex biological fluid like serum in the $<0.1\,\mu m$ range with excellent resolving power.

**Assessing intestinal permeability in mice with septic shock.** To further evaluate the potential of cFRAP-sizing, we wondered to which extent cFRAP could be suitable for a detailed assessment of the permeability of the intestinal barrier in mice (Fig. 1). The intestinal barrier is essential to prevent entry of the harmful intestinal content into the bloodstream. It consists of a single layer of epithelial cells that are sealed by tight junctions composed of claudins and other proteins in the junctional complex. Loss of intestinal barrier integrity is associated with various diseases, such as sepsis and inflammatory bowel disease. As there are indications that restoring intestinal integrity might ameliorate disease progression[22], there is great interest in finding pharmacological compounds, such as probiotics, that can safely strengthen the intestinal barrier. Therefore, methods are needed to accurately and quantitatively assess the intestinal barrier permeability. Inert fluorescently labelled dextrans of different molecular sizes $(3 - 2,000\,kDa)$ can be used for this purpose. A probe of a particular size is administered orally and the quantity that leaks into the blood is measured by fluorimetry, potentially coupled to

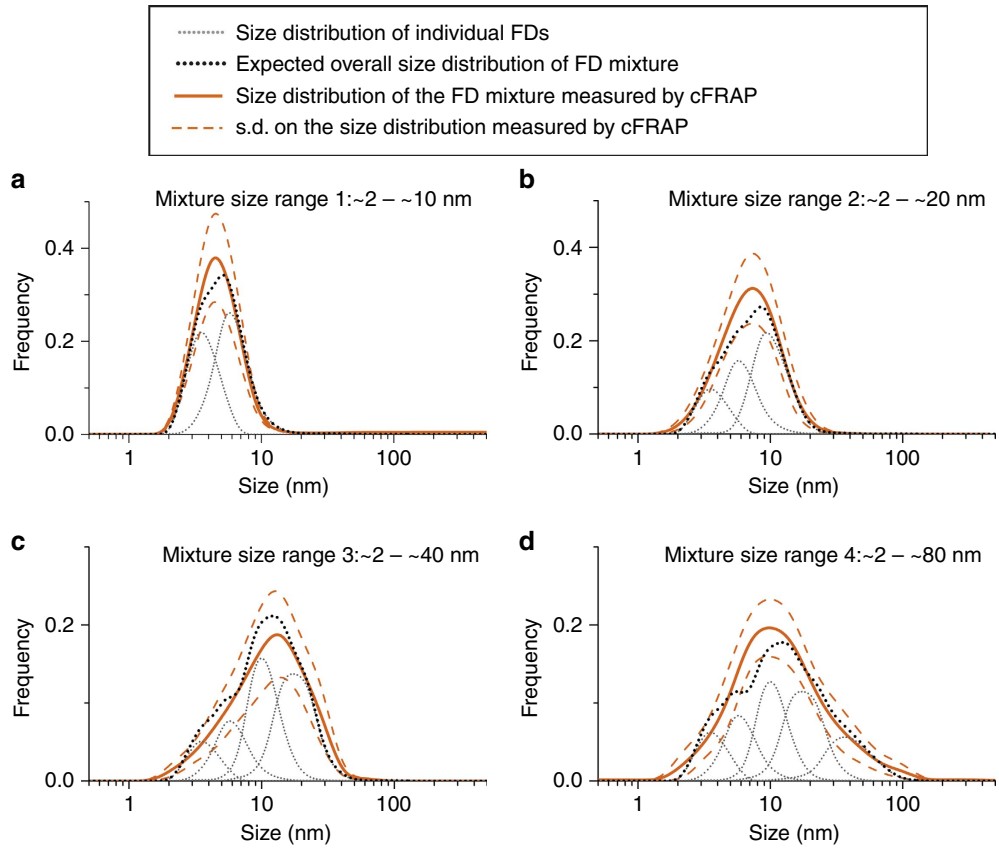

**Figure 2 | Experimental validation of cFRAP-sizing.** Mixtures were prepared with a gradually increasing number of FDs with different MW: (**a**) mixture of 1 mg ml$^{-1}$ FD4 (4 kDa FD) and 0.5 mg ml$^{-1}$ FD10 (10 kDa FD), (**b**) mixture of 1 mg ml$^{-1}$ FD4, 0.7 mg ml$^{-1}$ FD10 and 0.5 mg ml$^{-1}$ 40 kDa (40 kDa FD), (**c**) mixture of 1 mg ml$^{-1}$ FD4, 0.7 mg ml$^{-1}$ FD10, 0.3 mg ml$^{-1}$ FD40 and 0.3 mg ml$^{-1}$ FD150 (150 kDa FD), (**d**) mixture of 1 mg ml$^{-1}$ FD4, 0.6 mg ml$^{-1}$ FD10, 0.4 mg ml$^{-1}$ FD40, 0.25 mg ml$^{-1}$ FD150 and 0.25 mg ml$^{-1}$ FD500 (500 kDa FD). The grey dotted lines indicate the size distribution of the individual FDs measured by cFRAP. The black dotted lines are the sum of the grey dotted lines and represent the expected size distribution of the FD mixtures. The solid orange lines are the size distributions of the mixtures as experimentally measured by cFRAP, while the orange dashed lines indicate the s.d. ($n = 10$).

size exclusion chromatography[23]. To get information to which extent the barrier is compromised, this approach requires administering dextrans of various sizes sequentially—for which a new set of animals may be needed each time—and even then only discrete size information is obtained depending on the size of the probes used. In addition, these probes typically show some polydispersity so that the exact size that has leaked through is never certain.

To overcome these limitations, we propose the oral intake of a mixture of FDs covering a wide range of sizes. cFRAP can then be used to analyse the size distribution of FDs that have entered into the blood circulation after permeation through the intestinal barrier. A mixture of FD was prepared with a size ranging from ∼ 2–80 nm. We verified that the entire size distribution could be measured by cFRAP, both in PBS buffer and in serum (Fig. 4a). The FD mixture was administered by oral gavage to mice treated with an intraperitoneal injection of PBS (control) or lipopolysaccharides (LPS) to induce septic shock. As schematically shown in Fig. 4b, blood was collected by cardiac puncture, respectively, 7 and 20 h after induction of septic shock and plasma was prepared. In each case, oral gavage of FDs was done 5 h before blood collection. Plasma fluorescence was measured by fluorimetry to determine the overall fraction of intestinal FD that had entered into the blood circulation (Supplementary Fig. 11). These results confirm that septic shock results in loss of intestinal barrier integrity, as would be expected[24,25]. cFRAP-sizing was subsequently performed on the

plasma samples to determine the size distribution of FDs that had leaked through the epithelium (Fig. 4b). An exemplary cFRAP experiment is shown in Supplementary Fig. 12 and Supplementary Movie 1. To correct for differences in concentration and molecular brightness between the various FDs in the mixture, all size distributions in serum were normalized to the reference measurement of the FD mixture in PBS buffer (Fig. 4a). As for the control mice with intact intestinal barrier (treated with PBS only), the amount of FD in plasma was insufficient to perform meaningful cFRAP experiments. In LPS treated mice we found that at 7 h after LPS injection FDs in the lower size range (up to 10 nm) had entered into the blood circulation. Twenty hours after LPS injection the distribution had not substantially changed, which suggests that the barrier integrity does not change noticeably from 7 to 20 h after inducing septic shock. Note that subtle differences in the distributions may arise from the fact that the results at each time point are determined from a different set of animals.

To validate these results, we performed extra experiments according to the classical approach of administering FDs of different size separately in different mice and measuring the resulting fluorescence by fluorimetry in blood. In correspondence with the cFRAP experiments, we found that only FD with a nominal hydrodynamic size of ∼2.4 nm (FD4) or ∼4.0 nm (FD10) had entered the blood circulation, while larger FDs were not detected (Fig. 4c). Importantly, while ∼30 mice were needed to determine the size cut-off of the intestinal barrier by the

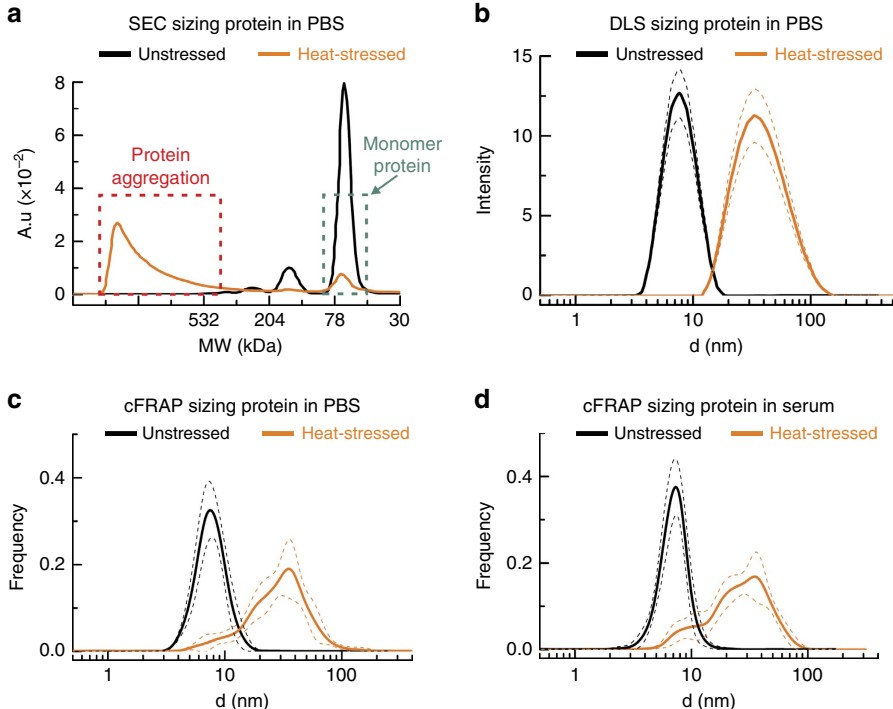

**Figure 3 | Sizing protein aggregates by cFRAP in comparison with SEC and DLS.** Aggregates of fluorescently labelled BSA were prepared through heat-stress. (**a–c**) FITC-labelled BSA in buffer is measured before (black) and after (orange) applying 6 h heat stress by (**a**) SEC, (**b**) DLS and (**c**) cFRAP. (**d**) Unstressed (black lines) and heat-stressed (orange lines) samples were diluted in (90%) serum. The solid lines show the size distributions as measured by cFRAP while the dashed lines indicate the s.d. on three independent repeats (with 10 cFRAP measurement per repeat). The size distributions as determined by cFRAP in serum nicely correspond to those in buffer. This demonstrates that cFRAP is very well capable of analyzing protein aggregates in (nearly undiluted) serum.

classical method of using different FD probes sequentially, microliter blood samples of only six mice were needed when cFRAP-sizing was used. This fivefold reduction in the number of lab animals is one of the great advantages of cFRAP-sizing to measure barrier leakiness *in vivo*.

**Assessing vascular permeability in mice with septic shock.** Vascular permeability is essential for supplying tissues with nutrients and clearing waste products. Vascular permeability may be increased by diseases such as inflammatory disorders and cancer, as well as by wound healing. This hyperpermeable state is believed to influence the composition of the extravasate and the pathways that solutes follow in crossing the vascular endothelium[26]. Vascular hyperpermeability may also affect the barriers in the brain, including the endothelial blood–brain (BBB) and epithelial blood–CSF barrier (BCSFB)[27]. Disruption of the integrity of the blood–brain barrier and blood–CSF barrier is believed to play a detrimental role in disease pathogenesis as protection of the delicate microenvironment of the brain from neurotoxic agents in the blood is compromised.

To quantify the size range of molecules that can leak through the vascular barrier, fluorescently labelled dextrans of different sizes can be injected intravenously, followed by analysis of fluorescence intensity in the relevant tissues (Fig. 1). According to the classic protocol, each size of dextran is to be injected separately, each time in a different animal. Instead, by intravenous administration of a mixture of FDs covering a broad range of sizes, here we demonstrate that a single experiment is sufficient when combined with cFRAP-sizing. First we confirmed that the full FD size range could be analysed in various organ fluids and CSF collected from control mice to which the FD mixture was added (Supplementary Fig. 13). Next, as

schematically shown in Fig. 5, mice were treated with an intraperitoneal injection of PBS (control) or LPS (septic shock). The mixture of FDs was injected intravenously 1 h before sample collection, which occurred, respectively, 7 and 20 h after PBS or LPS injection. CSF was collected using the cisterna magna puncture method. Kidney, brain, lung, spleen, ileum and liver were isolated after cardiac perfusion with PBS/heparin to remove all blood. Next, organ fluid was extracted from the different organs and total fluorescence was measured by fluorimetry after removing cellular debris, showing that there was a large increase in vascular permeability following septic shock (Supplementary Table 1). From 7 to 20 h, the permeability increased further for CSF, kidneys, lung and ileum. Next, these samples were analysed with cFRAP to determine the size distribution of FDs that had leaked through the vascular barrier. The size distributions (Fig. 5) were normalized to the relative fluorescence intensity as measured by fluorimetry (cfr. Supplementary Table 1) so that the y-axis reflects the amount of FD that has leaked through relative to the control (healthy PBS injected mice) for CSF or each organ. The cFRAP-sizing results are shown in Fig. 5 for CSF and kidney extract, while Supplementary Fig. 14 shows the cFRAP results obtained on extracts from brain, lungs, spleen, ileum and liver. Clearly, while no FD was found in the CSF of healthy control mice, FDs with a size below ∼10 nm significantly permeated from the blood into the CSF in LPS treated mice. In kidneys the endothelium was found to become more permeable over time with FDs up to ∼20 nm leaving the blood and entering the kidneys. Also here it is of note that 5 times less animals were needed as would have been the case for the classic fluorimetry method for which the size probes are to be administered separately in different animals. At the same time unprecedented detailed information is obtained on the continuous size range of probes that can leak through the barrier.

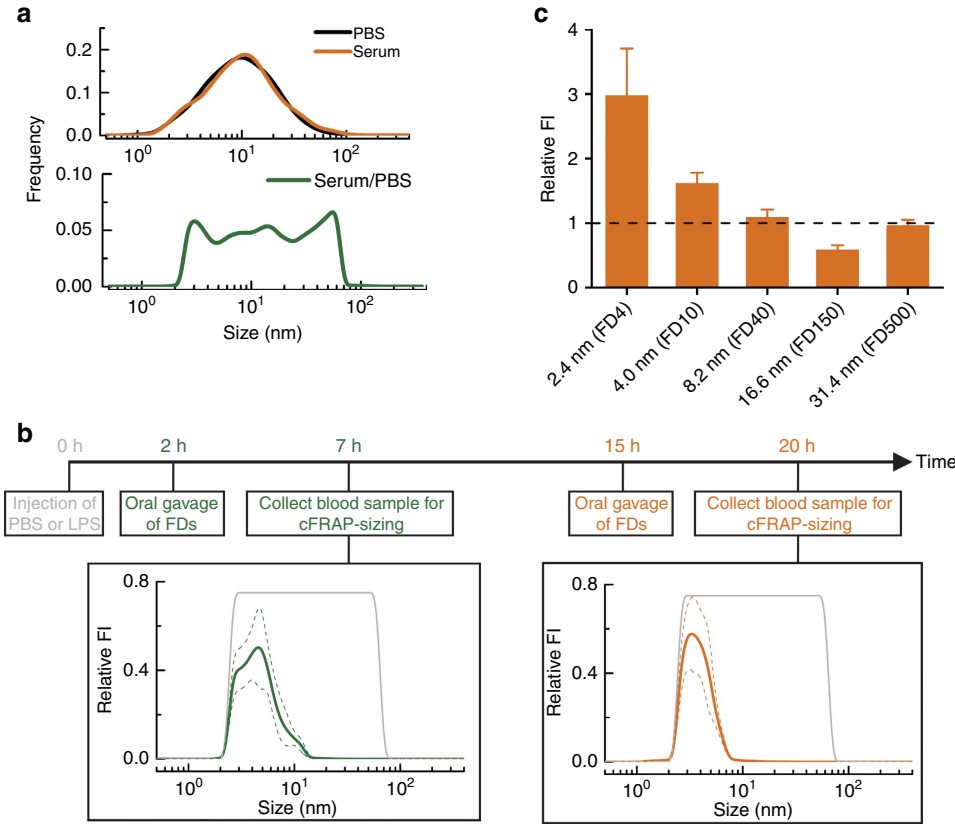

**Figure 4 | Assessing the permeability of the intestinal epithelium in mice.** To assess the permeability of the intestinal barriers, a mixture of FDs was prepared spanning a wide range of sizes from ~2 to ~80 nm. The mixture was prepared according to the weight ratio FD4:FD10:FD40:FD150:FD500 = 45:25:15:10:10. (**a**) The full-size range can be measured by cFRAP in PBS buffer and (95%) serum. By normalizing the serum data to the results in PBS, these differences between FD components are cancelled out (since $k_i$ is constant independent of the medium) and a virtually uniform distribution is obtained which is very well suited to interpret in a continuous fashion the size range of probes that can permeate through the barrier. (**b**) Following the induction of septic shock by intraperitoneal injection of LPS, a mixture of FDs covering a broad range of sizes (grey lines) was administered to mice by oral gavage, respectively, 2 and 15 h after LPS injection. Blood samples were collected, respectively, 7 h (green lines) and 20 h (orange lines) after LPS injection. Leakage of FDs through the intestinal epithelium in healthy mice (injected with PBS instead of LPS) was negligible and could not be measured by cFRAP. The data shown are average values obtained on 3 mice, with 10 cFRAP-sizing measurements per mouse. The solid lines are the average of all these results, while the dashed lines indicate the corresponding standard deviation. (**c**) To validate the cFRAP results on the intestinal barrier permeability, a classic experiment was performed where FDs of various sizes are administered separately to mice by oral gavage. The fluorescence intensity values are shown relative to the values of control mice (indicated by black dashed line). Only the values for FD4 and FD10 are significantly higher than the control case ($P < 0.05$).

## Discussion

FRAP has been used for decades to measure the average diffusion coefficient of fluorescently labelled molecules in various media, from cells and extracellular matrices to food products and drug delivery materials[28]. Instead, rather than measuring a single average diffusion coefficient, we succeeded in extracting the full distribution of diffusion coefficients from the recovery images using a dedicated theoretical framework that makes use of the full temporal and spatial information available in confocal recovery images. By doing so we achieve a substantially improved size resolution as explicitly demonstrated in Supplementary Fig. 2 and Supplementary Fig. 5, in comparison with 'standard' FRAP analysis where the fluorescence signal is integrated over the entire bleach area so that only time information on the recovery is taken into account[18]. As the cFRAP method is compatible with standard laser scanning confocal microscopes, it is easily accessible and straightforward to apply. Although cFRAP surely can be used to perform detailed biophysical diffusion studies, instead we have evaluated this technique to measure the size of nanomaterials in biological fluids. Considering the capabilities of typical confocal microscopes in terms of sensitivity and image

acquisition rate, cFRAP is perfectly suited to measure the diffusion of nanomaterials in the 1–100 nm size range in fluids. As such it nicely complements the fluorescence single particle tracking method that was recently developed in our group for nanoparticle sizing primarily in the 0.1–1 µm range[17,29].

A rectangular bleach area was chosen since for this geometry a full analytical solution is available that describes the recovery process both in time and space without any constraints on the size of the bleach by taking into account all the necessary parameters like the effective bleach resolution, the imaging resolution, and so on ref. 30. This model, originally developed for measuring single-component diffusion, was implemented here in the maximum entropy method (MEM) framework to extend its capability to analyse (semi-)continuous distributions of diffusion coefficients. By employing the full tempo-spatial information in confocal recovery images, we have shown that cFRAP offers better precision than techniques only using diffusion time information like DLS or the classic FRAP methods. Indeed, the PDI of size distributions of dextrans measured by cFRAP was significantly less compared with DLS. While also fluorescence correlation spectroscopy has been used to

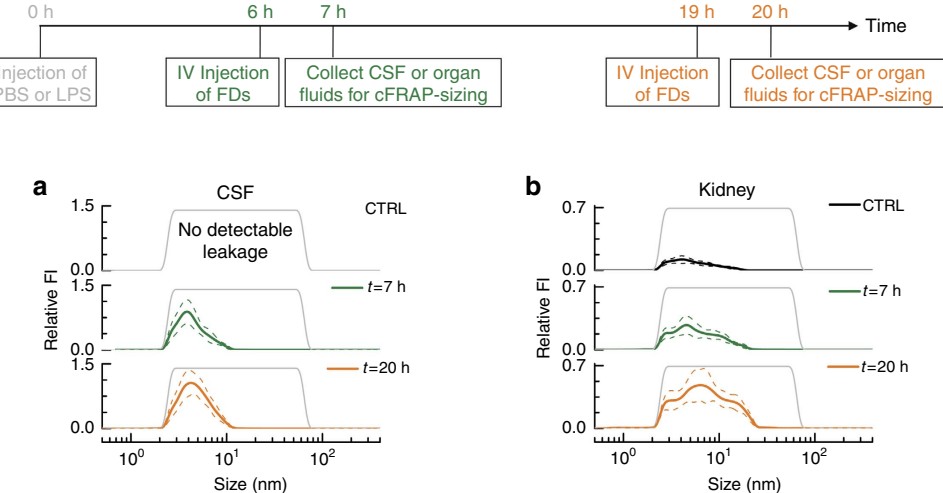

**Figure 5 | Assessing the vascular permeability in mice.** Following the induction of septic shock by intraperitoneal injection of LPS, a mixture of FDs covering a broad size range (grey line) was intravenously injected, respectively, 6 h and 19 h after the LPS treatment. CSF and organs were collected respectively at 7 h (green lines) and 20 h (orange lines) after the LPS treatment. cFRAP-sizing was performed on (**a**) CSF and (**b**) kidney-extract. Control mice were injected with PBS (instead of LPS) to determine the leakage in healthy mice as a reference (black lines). Note that in healthy mice FDs did not appear in the CSF. The data shown are average values obtained on 3 mice, with 10 cFRAP-sizing measurements per mouse. The solid lines are the average of all these results, while the dashed lines indicate the corresponding standard deviation. Note that the Relative FI values can only be compared for the various time points per fluid. Comparison of relative FI values should not be made between different sample types.

measure the size distribution of fluorescently labelled compounds, one can expect the same limited precision as for DLS as it also only deals with the time information from the diffusion process similar to DLS (ref. 31). When characterizing protein aggregation, we even found that a single cFRAP experiment produces the same size information as obtained by DLS and SEC combined. Furthermore, a major benefit of cFRAP is that microliter samples are sufficient. In principle even smaller volumes are very well possible since a single confocal image series typically probes a volume of 100 μm × 100 μm × 10 μm which corresponds to 0.1 nl. Assuming 10 measurements per sample this amounts to a probed volume of only 1 nl. Sizing by cFRAP, therefore, is perfectly compatible with miniaturization approaches like microfluidics.

As a first proof-of-concept application, we successfully demonstrated that cFRAP-sizing can be used to characterize protein aggregates directly in undiluted serum. This would be impossible to do by DLS due to strong light scattering by serum components. Also SEC is not without its problems since the high serum protein content may lead to interactions with the column matrix and altered elution profiles. Evidently the use of cFRAP comes at the expense of having to label the protein of interest. Yet, this is balanced by the fact that there are virtually no other techniques at this moment to characterize submicrometer protein aggregates in complex biological fluids.

Based on cFRAP-sizing we also devised a new approach to rapidly assess the integrity of the intestinal and vascular barriers which is related to disease pathogenesis. While disruption of tight junctions involved in cell − cell contact causes leakage of small molecules, the presence of epithelial apoptosis will allow permeation of larger molecules as well[32]. In the classic approach inert probes, such as FDs, of a particular size are administered to lab animals. However, as the sequential administration and fluorimetric analysis of probes of different sizes requires each time a different set of animals, it is time-consuming, expensive and poses ethical issues. Instead, we have demonstrated the use of a single mixture of probes (FDs) covering a very broad range of sizes. After sample collection of the relevant fluids, a single cFRAP experiment on a microliter sample can reveal the full-size distribution of probes that have permeated through the barrier. The cFRAP-sizing method, therefore, reveals in a single experiment the full-size distribution of probes that can leak through the barrier. Importantly, five times less animals were needed as compared with the classic fluorimetric method where FDs of (five) different sizes would have to be administered and analysed separately. Although there is one report that has tried to assess endothelial barrier permeability with a mixture of FITC dextrans and fluorescence SEC analysis[33], cFRAP-sizing has the clear advantage of being able to work with tiny sample volumes, even down to nanoliters if required. It is also very fast and does not require calibration (other than an intrinsic viscosity measurement) as is needed for SEC to interpret the elution profiles. Using cFRAP-sizing we even succeeded in analysing FD leakage in microliter samples of cerebrospinal fluid, notwithstanding that the fluorescence was very weak. To the best of our knowledge this would be impossible to do by SEC.

We conclude that sizing by cFRAP is a powerful and valuable new analytical technique for measuring the size of nanomaterials in complex biological fluids. While we have demonstrated its usefulness in proof-of-concept applications, the potential of cFRAP-sizing reaches much further and more valuable applications are expected to follow suit, potentially in combination with microfluidic approaches which enable accurate handling of small sample volumes.

## Methods

**Theory of cFRAP.** We start from the rectangle FRAP (rFRAP) model developed before for measuring a single average diffusion coefficient according to Deschout et al.,[30] which makes use of both time and spatial information in the recovery images:

$$F(x, y, t) = F_0 - F_0 \cdot \frac{K_0}{4} \cdot \left[ \text{erf}\left(\frac{x + \frac{l_x}{2}}{\sqrt{r^2 + 4Dt}}\right) - \text{erf}\left(\frac{x - \frac{l_x}{2}}{\sqrt{r^2 + 4Dt}}\right) \right]$$
$$\cdot \left[ \text{erf}\left(\frac{y + \frac{l_y}{2}}{\sqrt{r^2 + 4Dt}}\right) - \text{erf}\left(\frac{y - \frac{l_y}{2}}{\sqrt{r^2 + 4Dt}}\right) \right] \quad (1)$$

where $t$ is the time after photobleaching, $K_0$ the photobleaching parameter (which determines the extent of bleaching), $D$ is the isotropic diffusion coefficient of diffusing species, $l_x$ and $l_y$ are the width and height of the rectangular

photobleaching area, and $r^2$ is the mean square resolution of the bleaching and imaging point-spread function. In case of $N$ independent diffusing components, we can simply make a superposition of the individual fluorescence recovery profiles:

$$F(x, y, t) = \sum_i \alpha_i \varepsilon_i F_i(x, y, t) \qquad (2)$$

where $\alpha_i$ is the relative fraction of the $i$th component and $\varepsilon_i$ is the corresponding relative fluorescence brightness. Evidently, $\sum_{i=1}^{N} \alpha_i = 1$.

Defining:

$$k_i = \alpha_i \varepsilon_i K_{0i} \qquad (3)$$

the multicomponent rFRAP model becomes:

$$F(x, y, t) = F_0 - F_0 \cdot \sum_i \frac{k_i}{4} \cdot \left[ \mathrm{erf}\left( \frac{x + \frac{l_x}{2}}{\sqrt{r_i^2 + 4D_i t}} \right) - \mathrm{erf}\left( \frac{x - \frac{l_x}{2}}{\sqrt{r_i^2 + 4D_i t}} \right) \right] \\ \cdot \left[ \mathrm{erf}\left( \frac{y + \frac{l_y}{2}}{\sqrt{r_i^2 + 4D_i t}} \right) - \mathrm{erf}\left( \frac{y - \frac{l_y}{2}}{\sqrt{r_i^2 + 4D_i t}} \right) \right] \qquad (4)$$

The multicomponent rFRAP model of equation (4) can be generalized to describe a continuous distribution of diffusion coefficients $\alpha(D)$:

$$F(x, y, t) = \int \alpha(D) f(x, y, t, D, K_0(D), r(D)) dD \qquad (5)$$

where $f(x, y, t, D, K_0(D), \varepsilon(D), r(D))$ describes the fluorescence recovery of a component with diffusion coefficient $D$. Inserting equations (1) and (2) into equation (5) yields:

$$F(x, y, t) = F_0 - \frac{F_0}{4} \int \alpha(D) \epsilon(D) K_0(D) f'(x, y, t, D, r(D)) dD \qquad (6)$$

where $f'(x, y, t, D, r(D))$ is defined as:

$$f'(x, y, t, D, r(D)) = \left[ \mathrm{erf}\left( \frac{x + \frac{l_x}{2}}{\sqrt{r^2 + 4Dt}} \right) - \mathrm{erf}\left( \frac{x - \frac{l_x}{2}}{\sqrt{r^2 + 4Dt}} \right) \right] \\ \cdot \left[ \mathrm{erf}\left( \frac{y + \frac{l_y}{2}}{\sqrt{r^2 + 4Dt}} \right) - \mathrm{erf}\left( \frac{y - \frac{l_y}{2}}{\sqrt{r^2 + 4Dt}} \right) \right] \qquad (7)$$

For numerical computation according to the MEM we now make the transition to the semi-continuous case. Let $D$ be discretized in $n$ components (for example, with equal interval in $\log(D)$ space) in the range of $D_{\min}$ to $D_{\max}$, equation (7) becomes:

$$F(x, y, t) = F_0 - \frac{F_0}{4} \sum_{i=1}^{n} k_i f'(x, y, t, D_i, r_i) \qquad (8)$$

where we made use of equation (3). Equation (8) can be used for direct fitting to the pixel values in the recovery images.

Alternatively, the recovery data can be analysed based on the average intensities in ring areas (Supplementary Fig. 1b). This considerably reduces computation time while retaining the essential spatial information, as shown in Supplementary Note 1. The average intensity in ring $R_i$ is calculated as:

$$F_i(t) = \frac{1}{M_i} \sum_{(x,y) \epsilon R_i} F(x, y, t) \qquad (9)$$

where $F(x, y, t)$ is defined in equation (8) and $M_i$ is the number of pixels inside ring $R_i$.

Instead of performing a standard least-squares fitting of equation (9) to the experimental data, the MEM finds the 'best-fit' solution with maximum entropy. MEM ensures that the fitting result (that is, the distribution of diffusion coefficients) contains the least possible information to avoid over-interpretation of noise due to limited sampling statistics. In other words, it looks for the smoothest best-fit solution in the maximum entropy sense. The 'historic MEM' approach was implemented in this work, which means maximizing the Shannon-Jaynes entropy:

$$S = - \sum_{i=1}^{n} k_i \log k_i \qquad (10)$$

under the least-squares condition of $\chi^2 = M$, where $M$ is the total number of data points. For the pixel based fitting, the $\chi^2$ statistic is calculated by:

$$\chi^2 = \sum_i \sum_j \sum_k \frac{[F_{ij}(t_k) - F'_{ij}(t_k)]^2}{\sigma_{ij}^2(t_k)} \qquad (11)$$

where $F_{ij}(t_k)$ is the normalized fluorescence at position $x_i, y_j$ at time point $t_k$ and $\sigma_{ij}^2(t_k)$ is the corresponding variance. $F'_{ij}(t_k)$ is the corresponding theoretical value calculated from equation (8).

On the other hand, for the ring analysis, the $\chi^2$ statistic is calculated by:

$$\chi^2 = \sum_i \sum_k \frac{[F_i(t_k) - F'_i(t_k)]^2}{\sigma_i^2(t_k)} \qquad (12)$$

where $F_i(t_k)$ is the experimental average fluorescence in $i$th ring at time point $t_k$ and $\sigma_i^2(t_k)$ is the corresponding variance. $F'_i(t_k)$ is again the corresponding theoretical

value calculated from equation (9). The variance can be calculated for simulated images according to:

$$\sigma_i^2(t_k) = \frac{\sigma^2}{M_i} \qquad (13)$$

where $\sigma$ is the s.d. on the pixel values used for simulating the FRAP recovery images. For experimental images it can be calculated from ref. 34:

$$\sigma_i^2(t_k) = \frac{a F_i(t_k) + b}{M_i} \qquad (14)$$

Where $a$ and $b$ are constant parameters that can be determined by a series of images with various laser intensities of a homogeneous fluorescent solution with identical instrumental settings as in the final FRAP experiment[35].

Based on the theory outlined above, a Matlab code was written for MEM analysis of the recovery images which results in a semi-continuous distribution of diffusion coefficients, which can be converted to a distribution of sizes by the Stokes-Einstein equation when required. The Matlab code source is online available[36] and a user's guide is also available[37]. To ensure proper use of this method, it is important to stress two important experimental requirements. The theory is based on 2D diffusion only. In 3D extended samples (as is the case in this work) this means that bleaching should be performed with an objective lens of sufficiently low numerical aperture (typically <0.5) which produces a cylindrical laser beam in a substantial area above and below the focal plane. In that case, the bleaching will be quite uniform over an extended region along the optical axis, so that only 2D radial diffusion effectively takes place[38]. Second, in the derivation of the rFRAP model according to Deschout et al.[30], the assumption is made of a linear photobleaching process. In reality, however, photobleaching rather follows an exponential type of decrease. This means that the model will only work perfectly for modest bleach depths, that is, up to 50% photobleaching as demonstrated before[30]. This can be easily accommodated for by changing the bleach laser intensity appropriately.

**FRAP equipment and experimental procedure.** FRAP experiments were performed on a C1-si confocal microscope (Nikon, Japan) equipped with a 488 nm Ar-ion laser of 40 mW and acoustic optical tunable filter to modulate the laser intensity for bleaching and imaging (fastest imaging rate ∼0.5 frame per sec). Rectangular areas were photobleached and the fluorescence recovery was imaged using the Nikon NIS Elements AR software package. A ×10 numerical aperture 0.45 plan apochromat objective lens was used for bleaching and imaging. The laser power was adjusted to obtain 25–50% bleaching, in accordance with the theoretical requirement of limited bleaching (due to the assumption of a linear photobleaching process in the derivation of equation (1)). The recovery time depends on the diffusion coefficient as well as the size of the bleach area. A particular benefit of our theoretical framework is that we can adjust the size of the bleach rectangle in a continuous fashion. The smallest component in our applications is FD of 4 kDa (FD4), while the largest is FD of 500 kDa (FD500). To capture the diffusion from the smallest to largest components we used a bleach area of 50 μm with a sampling time that starts at 0.5 s per frame and increases to 16 s per frame towards the end of the time-lapse recording. As explained in detail in Supplementary Note 3 this ensures that the sampling was optimal over the entire experiment to capture the fastest and slowest component in one and the same measurement.

For FRAP experiments, 4 ul of the samples was 'sandwiched' between a microscope slide and a coverslip sealed by an adhesive spacer of 120 μm thickness (Secure-seal, Spacer, Molecular probes, Leiden, The Netherlands; Fig. 1). This provides a 3D environment for diffusion while avoiding flow in the sample. All FRAP measurements were performed at room temperature (22.5 °C).

**Simulation of FRAP images.** FRAP images were simulated using equation (2) in Matlab. Simulations were performed with the reported values of the diffusion coefficient(s) $D_i$, their relatively frequency $\alpha_i$, photobleaching amount $K_{0,i}$ and resolution parameter $r_i$.

**Viscosity measurement.** The measured distribution of diffusion coefficients can be converted to a corresponding distribution of hydrodynamic sizes (diameter), using the Stokes-Einstein equation $D = kT/3\pi\eta d$, where $k$ is the Boltzmann constant, $T$ the absolute temperature, $\eta$ the dynamic viscosity of the solution and $D$ the diffusion coefficient of the molecule. However, this requires accurate knowledge of the viscosity of the sample. While, for example, a capillary viscosity metre can be used for this in case of solutions prepared in the lab, it cannot be applied to the often minute samples retrieved from animal experiments. Therefore, we have made use of a viscosity probe with known size that can be added to the samples to inherently calibrate the viscosity of the sample under study. 10 kDa FD (FD10) at a weight concentration of 20 mg ml$^{-1}$ was added to the sample solution at a volume ratio of 1:20 so that the effect on the sample viscosity by adding the viscosity probe was considered neglectable.

**cFRAP data analysis.** Before fitting of the data to the cFRAP model, the recovery data (Supplementary Fig. 1) was normalized to the fluorescence before bleaching. Normalization to the pre-bleach intensity can be performed by dividing every pixel in the recovery images by the corresponding pixel in the pre-bleach image. To limit the corresponding amplification of noise, the pre-bleach image was

smoothed first with a median filter with a kernel of $5 \times 5$ pixels. Correction for laser fluctuations and bleaching during imaging is performed by dividing the pixels of each recovery image by the average value from one reference background region in the same image (Supplementary Fig. 1a). The reference background region was placed sufficiently far from the bleach region so as to remain unaffected by the diffusion front during the observation time.

Data analysis is done by fitting of the cFRAP model (equation (8)) to the pixel values of the normalized recovery images. Alternatively, as detailed in Supplementary Note 1, the region of interested can be divided into $n$ equally spaced ring-shaped areas. In that case equation (9) is fitted to the average intensity values in each of the ring-shaped areas. The MEM was included into the analysis so as to obtain the smoothest distribution fulfilling the requirement of $\chi^2 = M$, where $M$ is the total number of data points used for fitting. This is a well-known method to ensure that the final distribution does not contain more features than statistically warranted by the data[14]. In practice, using the function 'fmincon' in the Matlab Optimization tool box (The matworks, Natick, MA, USA) the entropy criterion according to equation (10) was maximized, while the constraint $\chi^2 = M$ in equation (12) was relaxed to the narrow interval $M - \sqrt{2M} \leq \chi^2 \leq M + \sqrt{2M}$.

**Covalent protein labelling with an extrinsic fluorophore.** A $10\,mg\,ml^{-1}$ solution of BSA was prepared by dissolving BSA lyophilized powder $\geq 96\%$ (Sigma-Aldrich) in carbonate buffer pH 8.3. The free amine groups of BSA were covalently labelled by 5–6-carboxyfluoresceine succinimidyl ester (5(6) FAM, SE; Life Technologies Corporation, Molecular Probes, Eugene, USA). For this purpose, $100\,\mu l$ of a $5\,mg\,ml^{-1}$ fluorescein solution in dimethylsulfoxide (Life Technologies Corporation, Ghent, Belgium) was added to a 2 ml $10\,mg\,ml^{-1}$ BSA solution and incubated for 1 h under constant gentle stirring. The incubation was stopped by adding $200\,\mu l$ of a $210\,mg\,ml^{-1}$ hydroxylamine (Sigma-Aldrich; stop solution) in ultrapure water adjusted to pH 8.5 with 4 M sodium hydroxide solution (Sigma-Aldrich). Next, the excess of free fluorescein labels was removed by dialysis overnight against 4 L 0.1 M phosphate pH 7.0 in a Slide-a-lyser 20 kDa MWCO (Thermo Scientific, Rockford, USA). The phosphate buffer was adjusted to pH by varying the amount of 0.1 M monobasic dihydrogen phosphate (WR, Leuven, Belgium) and 0.1 M dibasic monohydrogen phosphate (Merck, Darmstadt, Germany) solution. The buffer was filtered through a $0.2\,\mu m$ PES filter (Novolab, Geraardsbergen, Belgium) before use.

Soluble protein concentration and labelling efficiency was determined by measuring the absorbance at 280 nm and at 495 nm respectively on a Spectramax M2 (Molecular devices, Sunnyvale, USA) with the SoftMax pro software version 6.1. For all measurements, samples were diluted 10-fold and $200\,\mu l$ samples and buffer controls were transferred in triplicate to a 96-well plate (96-well pureGrade, non-sterile, transparent, F-bottom, Novolab).

The degree of labelling was calculated using the measured absorbance of the dye at its absorbance maximum of 495 nm (blank corrected) and according to the following equation:

$$DOL = \frac{A_{maxdye} \times MW_{protein}}{C_{protein} \times \varepsilon_{dye}}$$

with $MW_{protein}$ for the molecular weight of the protein, $\varepsilon_{dye}$ for the molar extinction coefficient of the dye ($68000\,cm^{-1}\,M^{-1}$) at its absorbance maximum (494 nm) and $C_{protein}$ is the protein concentration ($mg\,ml^{-1}$). The Lambert-Beer law is used to calculate the protein concentration. For this application, the measured protein absorbance at 280 nm was corrected for the dye absorbance at 280 nm according to manufactures' instructions using the following equation: $A_{protein} = A_{280\,nm} - A_{max\,dye}(CF)$ with $CF = \frac{A_{280\,freedye}}{A_{max\,freedye}}$.

**Dynamic light scattering size measurements.** DLS size measurements were performed on a Malvern Zetasizer Nano ZS (Malvern Instruments Ltd., Malvern, UK) equipped with a 632 nm, 4 mW He-Ne laser source. Instrument performance was verified by a system suitability test according to the manufacturer's IQ/OQ documentation and consisting of a measurement of 60 and 200 nm polystyrene beads from Thermo Scientific (Erembodegem, Belgium). For each measurement, $40\,\mu l$ samples were transferred into a ZEN0040 Micro cuvette (Malvern Instruments Ltd.). Sample measurements were performed at $25\,°C$ with automatic attenuation. Samples were equilibrated for 180 s at $25\,°C$. Measurement position was fixed at 4.65 by 'The seek for optimum position' option. The sample was measured 10 times using automatic measurement duration with a delay of 60 s between each measurement. Measurements were performed under a $173°$ backscattering angle. For data processing, the general purpose algorithm was used. The PDI was defined as $PDI = (\sigma/d)^2$, where $\sigma$ is the s.d. of the size distribution and $d$ the mean diameter.

**Size exclusion chromatographic measurements.** The protein size distribution in the non-stressed and stressed samples for labelled and non-labelled BSA was further evaluated by size exclusion chromatography. For chromatographic separation, a Yarra SEC-3000 column ($300\,mm \times 4.6\,mm \times 3.0\,\mu m$) attached to a Security cartridge GFC 3000 ($4 \times 3.0\,mm$) (Phenomenex, Utrecht, The Netherlands) was installed on an Acquity H-Class UPLC BioSystem (Waters, Milford, MA, USA) equipped with a PDA detector with a 5 mm 1500 nl titanium flow cell. Empower 2 was used as operating system. The mobile phase consisted of 0.1 M phosphate pH 7.0 and the used Gel Filtration Standard was from Bio-Rad (Temse, Belgium). Before injection of the sample and standards on the column, the insoluble aggregates were removed by centrifugation in a 5424R centrifuge with FA-45-24-11 rotor (Eppendorf) for 10 min at 15,000 r.p.m. The concentration of the soluble protein was determined on a spectramax M2 multi-detection reader using the Lambert-Beer equation with the measured absorbance at 280 nm and the theoretical absorbance at 280 nm for a 1% solution (being 0.66 for BSA). In addition, the absorbance at 320 nm was measured for background correction. During analysis, the autosample tray and column oven sample tray were both thermostated at $22\,°C$. The flow rate was set on $0.35\,ml\,min^{-1}$. After equilibration with the mobile phase, samples and standards were injected in triplicate. $20\,\mu g$ was injected for each sample. Protein elution from the column was detected at 280 nm, while 320 nm was recorded as background control. The molecular weight standard was injected in triplicate. For the proteins within the molecular weight separation range of the column, a linear correlation between logarithm of the molecular weight and the elution time was established and used for size estimation of the aggregated and non-aggregated proteins.

**Temperature stressed protein.** Labelled and non-labelled BSA was diluted 10-fold to obtain a $1\,mg\,ml^{-1}$ solution. Next, the sample was distributed over 1.5 ml Eppendorf tubes (Eppendorf, Hamburg, Germany) in 1 ml fractions and subjected to temperature stress on incubation in a Thermomixer comfort (Eppendorf) for 6 h at $75\,°C$. A non-stressed labelled and non-labelled sample was retained at $4\,°C$. During all manipulations over the different steps, the samples were kept protected from light.

To measure the size distribution of protein in biological fluids, blood was withdrawn on citrate from healthy volunteers under informant consent. Plasma was prepared by centrifugation. For analysis of proteins with the cFRAP method, mixtures of plasma and protein (90/10, v/v plasma/protein) were prepared. The same dilutions of protein in buffer were made to serve as a control.

**Fluorescent/dextran probes.** FD or dextran (D) of various molecular weight (MW) (FD4/D4: $MW = 4 \times 10^3\,g\,mol^{-1}$, FD10/D10: $MW = 1 \times 10^4\,g\,mol^{-1}$, FD40/D40: $MW = 4 \times 10^4\,g\,mol^{-1}$, FD150/D150: $MW = 1.5 \times 10^5\,g\,mol^{-1}$, FD500/D500: $MW = 5 \times 10^5\,g\,mol^{-1}$) were purchased from Sigma-Adrich (Bornem, Belgium). For the validation experiments of cFRAP, FD solutions were prepared in HEPES buffer at pH 7.0 and dextran solutions were prepared in distilled water. The concentration was always $0.5\,mg\,ml^{-1}$ for DLS measurements. For each type of FD, a concentration series was prepared to determine the linear fluorescence range as observed on the confocal microscope. For the *in vivo* permeability measurements, different FDs were mixed at a weight ratio (FD4:FD10:FD40:FD150: FD500 = 40:25:15:10:10) and subsequently dissolved in PBS buffer, where the concentration of FD4 was $80\,mg\,ml^{-1}$. While this is outside the linear fluorescence range, such a high concentration was chosen to compensate for the dilution that occurs when the mixture is injected into mice.

**Animals.** C57BL/6J mice were housed in an SPF animal facility with *ad libitum* access to food and water. Both male and female mice (8–12 weeks old) were used. All experiments were approved by the ethics committee of the Faculty of Science of Ghent University. Mice were distributed randomly in different cages and mice from the same cage were randomly allocated to different experimental groups. They were injected intraperitoneally (i.p.) with $8.75\,mg\,kg^{-1}$ body weight LPS from *Salmonella enterica* serotype abortus equi (Sigma), an $LD_{100}$ dose for wild-type C57BL/6J mice. No statistical method was used for sample size estimate.

***In vivo* experiments on intestinal permeability.** Control mice (injected with D-PBS) were sampled 7 h after injection. Septic shock mice were sampled 7 and 20 h after induction of peripheral inflammation. FITC-labelled dextran solution was administered to mice by gavage 5 h before sampling. Blood obtained by heart puncture was collected in EDTA-coated tubes (Sarstedt) and plasma was prepared. Leakage of FITC-labelled dextran into the circulation was determined by measurement of the fluorescence with $\lambda_{ex}/\lambda_{em} = 488/520\,nm$. Values were normalized to the PBS control value. After fluorescence measurement, these samples were also measured by cFRAP in a similar fashion as for the vascular permeability experiments. No blinding was done for all of samples. The fluorescence intensity of these samples as measured by fluorimetry were compared by one-way analysis of variance. No samples or animals were excluded from the analysis

***In vivo* experiments on vascular permeability.** Control mice (injected with D-PBS) were sampled 7 h after injection. Septic shock mice were sampled 7 and 20 h after induction of peripheral inflammation. One hour before sampling, mice were injected intravenously (IV) with the relevant FITC-labelled dextran solutions. CSF was harvested from the fourth ventricle, centrifuged at $300g$ to remove all cell debris, and cerebrospinal fluid (CSF) supernatant was collected and diluted 50-fold in D-PBS (Invitrogen) before analysis. Next, mice were transcardially perfused with D-PBS supplemented with heparin to remove all labelled dextran in circulation. Organs were isolated, cut into small pieces and incubated with formamide to extract the remaining FITC-labelled dextran from the tissues. After overnight

incubation at 37 °C, samples were centrifuged and supernatant was collected. Finally, the fluorescence of CSF and organ supernatant was measured at $\lambda_{ex}/\lambda_{em} = 488/520$ nm by Fluostar Omega and values were normalized to the PBS control per tissue. Again, the samples of the fluorescence were compared by one-way analysis of variance. The samples were subsequently analysed by cFRAP to determine the size distribution of FDs in the various bodily fluids and organs.

**Data availability.** Data supporting the findings of this study are available within the article and its Supplementary Information Files. The Matlab source code and a user guide are available for download[36,37].

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

## Acknowledgements

Financial support by the Ghent University Special Research Fund (Centre for Nano- and Biophotonics) is acknowledged with gratitude. R.X. gratefully acknowledges the financial support from China Scholarship Council (CSC). K.B. and R.E.V. would like to acknowledge the Research Foundation Flanders (FWO Vlaanderen). R.E.V. also acknowledges the Interuniversity Attraction Poles Program of the Belgian Science Policy (IAP-VI-18). K.B. acknowledges financial support from the European Research Council (ERC) under the European Union's Horizon 2020 research and innovation programme (grant agreement no 648124).

## Author contributions

K.B. had the idea for the cFRAP methodology and its application for protein sizing. S.C.D.S. had the idea for assessing barrier permeability by cFRAP. R.X. developed the cFRAP methodology and code, and performed all cFRAP measurements and data analysis. R.V.B. and E.V.W. performed all *in vivo* experiments. K.B. prepared protein samples and performed DLS and SEC measurements. J.D., S.C.D.S., C.L. and K.B. advised on experiments, data analysis and paper writing. All authors discussed the experimental results and wrote the paper.

## Additional information

**Competing financial interests:** The authors declare no competing financial interests.

