## [Peer Review File · Nature Communications]

Reviewers' comments:

Reviewer #1 (Remarks to the Author):

In this paper a new method for utilizing FRAP measurements to assess the size distribution of fluorescently labelled particles in physiologic fluids (and likely other situations) is outlined. The new approach was extensively characterized and validated, and demonstrated to provide possible advantages over existing methods. The paper was well written and logically presented. Given the cost of confocal microscopes and FRAP setups, one might wonder how widely adopted the technique could be, however, it clearly has potential applications. I had only one comment: on page 5, the authors demonstrate the ability to assess protein aggregation in serum. It would be useful for them to comment on the concentration limitation of the technique, as oftentimes, therapeutic proteins are present in very low concentrations.

Reviewer #2 (Remarks to the Author):

This paper reports a new analytical method to measure the size distribution of fluorescent nanoobjects in blood samples using fluorescence recovery after photobleaching (FRAP). The authors have demonstrated its applicability for assessment of therapeutic proteins and the permeability of the intestinal and vascular barriers of mice. Diffusion coefficients from fluorescent species are converted to their radii using the Stokes-Einstein equation. The use of fluorescence technique helps selectively obtain size information with minimum influence of light scattering, which hinders measurements in complex solutions such as blood samples. The size distribution from the FRAP measurements is compared with those obtained using conventional techniques such as dynamic light scattering and size-exclusion chromatography.

Unfortunately, the paper did not cite and discuss any closely-related papers that have determined the size distribution of nanoobjects using FRAP (for example, Periasamy/Verkman, *Biophys. J.* 1998, 75, 557.; Hauser/Seiffert/Oppermann, *J. Microsc.* 2007, 230, 353.; Tycon/Daddysman/Fecko, *J. Phys. Chem. B* 2014, 118, 423.). These previous papers have employed very similar theoretical models for the same purpose. The methodological differences between this and previous works are the use of photobleached areas of different shapes. However, the paper does not describe why the rectangular photobleaching is required instead of common circular one. The analysis based on rectangular rings is probably new, but similar analysis based on radial circular rings is more straightforward for isotropic diffusion systems. Note that the *J. Phys. Chem.* paper has demonstrated in vivo FRAP measurements of GFPs in living cells.

In addition, the paper only provides size distribution graphs (e.g., Figures 2-5) that are the final outcome of the analysis, and does not provide any raw images and graphs that are required to assess the data analysis method. In particular, those from blood samples need to be shown, as the readers of this paper should be curious about the S/N for solutions containing interfering species. In general, FRAP needs fluorescence species of μM or sub- μM range, and it is interesting that the blood samples directly obtained from the animals contain FDs of the relatively high concentrations.

Furthermore, how long does it take to record one image using the scanning confocal microscope? The image acquisition time must be sufficiently short to measure the fast diffusion from small nanoscale objects.

Describe the scientific reasons behind the narrower size distribution from FRAP than from DLS (page 5, 2nd paragraph and page 6, 1st paragraph). It is unclear how the spatial information in the FRAP approach leads to the improved determination of size distribution. This issue is very important in order to compare FRAP with FCS (page 10, 3rd paragraph). The fluorescence-based selectivity is common to FRAP and FCS.

In Figure 3ab, the heat-stressed samples give peaks/shoulders at d significantly smaller than those of unstressed samples. Explain what species may give the peaks/shoulders? In Figure 4, what causes the difference in the size distribution between 7 h and 20 h samples? The distributions are clearly different though the authors claim that they are similar (page 8, 1st paragraph).

In conclusion, I have reservations about the suitability of this paper for publication in Nature Communications due to its limited novelty and insufficient scientific description/discussion. This is a rather technical paper, and should be submitted to a more specialized journal. It is important to clearly describe the scientific (and technical) novelty of the method in the main text.

Reviewer #3 (Remarks to the Author):

The manuscript by Xiong et al presents a novel approach to fluorescence recovery after photobleaching, which allows to fit the recovery profile to a distribution of diffusion coefficients, i.e. a polydisperse or multi-component sample. This is applied to determine aggregation sizes in protein samples and size distributions of molecular species transferring through vascular or intestinal barriers. While I would not consider myself an expert in these particular application fields, I think these are very relevant and the data seems very convincing with considerable checks and balances. All in all, I find this a very thorough work with detailed explanations and additional analyses in the supplemental work. Also, I think there is a majority of other applications that could be opened up with this cFRAP method, on which the authors could expand a bit more in the discussion (anyway, I find the discussion reads more like an extensive summary of the results section than as a discussion; this is one thing the authors could improve upon) Intriguing would also be application to spatially varying diffusion coefficients.

There is an extensive body of supplemental work along with the manuscript. Both are very well readable, and in the initial part of the paper it is easily combined reading through both, later in the presentation of results in Figures 3 and 4, the reader has to go forth and back between the two, and also between different supplemental figures, a lot, which considerably hinders reading. Can't supplemental figure 8a+b be included in figure 3? And sfig 9 and 11 in fig.4, or somehow else to improve reading (9c,d are only discussed after sfig 11)?

The acronym rFRAP (rectangular FRAP) is used without specification, both in manuscript (p17), as in supplement (note 1)

Supp. note 1: 'in order to speed up the fitting...': shouldn't this also improve the fitting by improving the signal to noise with a factor $\sqrt{\# \text{pixels in rectangle}}$? Or are integration times such that single-pixels noise levels are not a limiting factor in the fits?

Supp note 4: The monodisperse single and double component distribution fittings already give an apparent polydispersity just due to the fitting uncertainties. So when is this polydispersity real and how can this be assessed? Is there a goodness of fit, or model test parameter that can be used on the experimental and MEM-fitted recovery results?

caption SFig 1: ' $t=1.6, 10.6, \text{ and } 50.6 \text{ s}$ ', latter should be 6.4 and 33.4 according to figure.

SFig4 e,f: can these be compared by plotting it together with results for $T=\tau_{\text{slow}}$ and $\Delta t=0.5\tau_{\text{fast}}$?

On the cFRAP to DLS comparison: are the concentrations the same? DLS needs very dilute suspensions (0.5mg/ml) according to Methods. It is not clear what the cFRAP concentrations are, I only could find stated 80mg/ml for the solution before injection into the animals. FRAP requires

sufficient labelled species in the bleaching area, so do they measure the same diffusion coefficients or do we already see a concentration dependent correction for the FRAP result. Different concentration ranges that can be addressed could also be discussed in the discussion.

Sfig 9c, d: please use same numbering for same fluids in c and d. In legend for c ileum is misspelled.

Captions Sfig 10 and 11: shock is misspelled

Page 10, discussion: on cFRAP vs DLS: incorporating the full spatial information in cFRAP improves the FRAP result itself, or maybe better it removes the error made by not including it. I do not see why this improves it compared to DLS, this is more a matter of averaging between both techniques. And maybe probing slightly different diffusion coefficients, free self diffusion in DLS and with concentration-dependent correction in FRAP.

Response to the referees' comments

Reviewer #1

We would like to thank the reviewer for his/her overall appreciation of our manuscript. Please find our response to your comment below.

1. **"I had only one comment: on page 5, the authors demonstrate the ability to assess protein aggregation in serum. It would be useful for them to comment on the concentration limitation of the technique, as oftentimes, therapeutic proteins are present in very low concentrations."**

The lower limit of detection essentially depends on the sensitivity of the confocal microscope and the fluorescence brightness of the labelled molecules, i.e. the proteins in this example. At low concentrations (i.e. low fluorescence intensity) the detector gain typically needs to be increased, which will result in more noise and, therefore, a lower signal to noise ratio. At a certain point the SNR will become too low to perform meaningful analysis.

As this is a fundamental issue, we decided to perform additional experiments and simulations to see what is the lower SNR limit for which cFRAP still can be used. First we prepared a series of FD40 solutions with gradually decreasing concentration in which cFRAP experiments were performed. As the concentration goes down, the SNR decreases as well, as can be seen in the newly added Supplementary Figure 7. Down to SNR = 2.4 cFRAP analysis could be correctly performed, corresponding to a FD40 concentration of 4 $\mu\text{g/ml}$ or 100 nM. In addition, we created simulated cFRAP images with similarly decreasing SNR which confirmed almost exactly the experimental results with FD40 (also see Supplementary Figure 7).

Next, to answer the question by the reviewer more precisely, we performed similar experiments on a dilution series of labelled BSA. As the results from Supplementary Figure 10 show, cFRAP analysis was possible down to a concentration of 4 $\mu\text{g/ml}$ (~60 nM).

We would like to stress again that the lower concentration limits mentioned here are dependent on the type of confocal microscope used and the brightness of the labelled molecules. The SNR is the more fundamental criterion in respect, which is why we reported those values in Supplementary Figure 7.

We added this information to the 'Validation of cFRAP-sizing' section in Results (page 5), where reference is made to the new data in Supplementary Figure 7. The lower concentration limit for BSA was mentioned in the section 'Characterization of protein aggregation in biological fluids' (page 6) with reference to Supplementary Figure 10.

Reviewer #2

We thank the reviewer for critical reading of our manuscript. We have carefully considered the comments that were raised and give a detailed response per question below. We also indicate how the manuscript was adjusted.

1. **“Unfortunately, the paper did not cite and discuss any closely-related papers that have determined the size distribution of nanoobjects using FRAP (for example, Periasamy/Verkman, *Biophys. J.* 1998, 75, 557.; Hauser/Seiffert/Oppermann, *J. Microsc.* 2007, 230, 353.; Tycon/Daddysman/Fecko, *J. Phys. Chem. B* 2014, 118, 423.). These previous papers have employed very similar theoretical models for the same purpose. The methodological differences between this and previous works are the use of photobleached areas of different shapes. However, the paper does not describe why the rectangular photobleaching is required instead of common circular one. The analysis based on rectangular rings is probably new, but similar analysis based on radial circular rings is more straightforward for isotropic diffusion systems. Note that the *J. Phys. Chem.* paper has demonstrated *in vivo* FRAP measurements of GFPs in living cells.”**

Thank you for bringing those articles to our attention. We are very much familiar with the article by Verkman and Periasamy. We planned to make reference to this article in our manuscript, but it seems that in a final editing phase the reference got mistakenly deleted, unfortunately (Note that the article is referenced in a recent FRAP review of ours, Loren et al. *Quarterly Reviews of Biophysics* 48, 2015). Verkman & Periasamy have shown that MEM analysis allows to extract a (semi)continuous distribution of diffusion coefficients from FRAP recovery curves. However, they did not use it for size measurements of nanomaterials, and certainly not for sizing in biological fluids. On a technical level, the improvement of our method over the one reported by Verkman & Periasamy, is that we not only analyze the fluorescence recovery over time, but additionally include spatial analysis. By doing so we achieve a substantially improved size resolution as can be seen in Supplementary Fig. 2. The case $NC=1$ corresponds to 'standard' FRAP analysis where only the time-progression of the mean fluorescence in the bleach area is considered. Instead, by performing spatial analysis over time, the resolution of the distribution of diffusion coefficients is clearly improved (e.g.

NC=10 or NC=20). As it is an important aspect in our work, we have decided to add extra data in the revised manuscript to show this point more explicitly. In Supplementary Figure 5 we have investigated the ability of cFRAP vs standard FRAP to discriminate two subpopulations. We find that standard temporal analysis (as in Verkman & Periasamy) can only resolve two populations if their diffusion coefficient differs by a factor of 8. Instead, by including spatial analysis we find that cFRAP can still resolve both populations if they differ by a factor of 3 only. Therefore, thanks to including spatial information into the analysis, the resolution is improved by a factor of 3. This issue was discussed in the main text in section 'Validation of cFRAP-sizing' on page 5.

The article by Tycon, Daddysman and Fecko was unknown to us, so thank you for pointing it out. It is similar in approach to that of Verkman and Periasamy in the sense that also here only time information is considered. The method is also limited to spot photobleaching measurements, while our approach allows continuous flexibility in choosing the size of the bleach area. This is important to optimize recovery time vs diffusion rate (recovery time scales with the square of the bleach area of a given diffusion coefficient). The article is nevertheless relevant as it discusses measuring the size of molecular complexes in cell lysate.

The article by Hauser, Seiffert and Oppermann was known to us. These authors derive an expression for multi-component diffusion and also implement MEM analysis to obtain (semi)continuous distribution of diffusion coefficients. The experimental validation, however, is limited to two-component fitting accompanied by a statement that MEM could not resolve the two components. The reviewer will appreciate that even just in our supplementary material we went much further and really show in detail the capability and limitations of MEM analysis to resolve multiple components and to analyze systems with a continuous distribution of diffusion coefficients. In particular, with cFRAP we can resolve components that differ by a factor of 3 only (cfr. Suppl. Fig. 5), which was unsuccessful in the Hauser article. In any case, the method by Hauser et al was not used to determine the size of nanomaterials in biological fluids. On a more technical note, the equations used in the work by Hauser et al do not take into account the fact that FRAP images are observed through a microscope with finite resolution. While it may not matter for large bleach areas, but for smaller ones it can make a noticeable difference [Braeckmans et al. Biophys Journal 2003]. Due to this theoretical 'incompleteness' and the limited experimental validation we did not reference the article by Hauser et al. in the first version of our manuscript.

In the revised version we now make clear reference to all three articles in the introduction. A new Supplementary Note 4 was added in which the issue of resolution is investigated and described in detail. Reference is made to Supp. Note 4 in the main text in the section 'Validation of cFRAP-sizing' on page 5.

- 2. "In addition, the paper only provides size distribution graphs (e.g., Figures 2-5) that are the final outcome of the analysis, and does not provide any raw images and graphs that are required to assess the data analysis method. In particular, those from blood samples need**

to be shown, as the readers of this paper should be curious about the S/N for solutions containing interfering species. In general, FRAP needs fluorescence species of μM or sub- μM range, and it is interesting that the blood samples directly obtained from the animals contain FDs of the relatively high concentrations."

The reviewer will appreciate that the amount of data (confocal time-lapse series) behind these graphs is huge. Making many GB of raw data available as supplementary information seems a step too far. Instead, in the revised manuscript we have now included a representative experiment that was performed in blood. The original time-lapse movie is provided as Suppl. Movie 1 and in Suppl. Fig. 12 we show details of the analysis.

The concentration of dextrans applied into the animals was optimized (FD4:FD10:FD40:FD150:FD500=40:25:15:10:10, with 80 mg/ml FD40) so that the final concentration in blood and organ extracts was sufficient to perform cFRAP analysis. This, of course, initially required some optimization but finally it seemed that the concentrations that we used were suitable to perform cFRAP analysis. Also see our response to Reviewer 1 with regard to the lowest measurable concentration.

- 3. "Furthermore, how long does it take to record one image using the scanning confocal microscope? The image acquisition time must be sufficiently short to measure the fast diffusion from small nanoscale objects."**

The recovery time depends on the diffusion coefficient, as the reviewer points out, but also on the size of the bleach area. A particular benefit of our theoretical framework is that we can adjust the size of the bleach rectangle in a continuous fashion as we explicitly take the effect of the bleaching and imaging point spread function into account. Some numbers: The fastest imaging rate of our particular laser scanning confocal microscope is ~ 0.5 s/frame (512x512 pix). The smallest FITC-dextran (FD4) is 4 kDa, which has a diffusion coefficient in serum of $\sim 120 \mu\text{m}^2/\text{s}$. To accurately capture the diffusion of such fast components we used a bleach area of $50 \times 50 \mu\text{m}^2$. Under these conditions the characteristic recovery time is ~ 1.3 s ($\tau = (l/2)^2/4D$, where l is the length of one side of the bleach rectangle). In the first part of the validation of our method we have studied what is the minimal sampling needed to accurately detect a certain diffusion coefficient (please see Supplementary Note 3). Here we found that the time interval should be smaller than or equal to half of the characteristic recovery time. This means that the sampling time should be 0.65s or less to capture the diffusion of FD4 in serum. As our frame rate was 0.5 s/frame we were in the right condition to measure the fastest component in our system.

We have further clarified this point in the Methods section 'FRAP equipment and experimental procedure' on page 18.

- 4. "Describe the scientific reasons behind the narrower size distribution from FRAP than from DLS (page 5, 2nd paragraph and page 6, 1st paragraph). It is unclear how the spatial information in the FRAP approach leads to the improved determination of size**

distribution. This issue is very important in order to compare FRAP with FCS (page 10, 3rd paragraph). The fluorescence-based selectivity is common to FRAP and FCS.”

DLS and FCS are based on fluorescence time measurements in a focused laser beam. This is, from conceptual point of view, similar to what was done by Verkman and Periasamy. Also there the fluorescence is measured over time in the spot bleached by a focused laser. By including not only time, but also spatial information, much more ‘subtle’ differences in the recovery can be ‘sensed’, such as those coming from multiple diffusing species. This was in essence also the concept on which the work by Hauser et al. (see comment 1) was based. The gain in resolution by including spatial analysis can be nicely seen in Supplementary Figure 2. Here, the condition NC=1 corresponds to integrating the signal over the entire area and analyzing the fluorescence recovery over time (similar to Verkman and Periasamy, or Tycon et al). Also see the explanation that was given at the end of Suppl. Note 1. By including spatial information into the analysis (cfr. increasing NC values in Suppl. Fig. 2), the apparent width of the distribution (coming from a single component) decreases, i.e. the resolution increases.

To stress this essential fact further we have added simulations in the revised manuscript showing that full spatio-temporal analysis with cFRAP can discriminate two components that differ only a factor of 3, while this is a factor of 8 if only time information is taken into account (Suppl. Fig. 5).

- 5. “In Figure 3ab, the heat-stressed samples give peaks/shoulders at d significantly smaller than those of unstressed samples. Explain what species may give the peaks/shoulders? In Figure 4, what causes the difference in the size distribution between 7 h and 20 h samples? The distributions are clearly different though the authors claim that they are similar (page 8, 1st paragraph).”**

It is an interesting remark. The original data was based on 10 technical repeats from the same batch of aggregated proteins. To ascertain that those subtle features are really there, **we performed these experiments again (3 entirely independent repeats) and combined those data in the determination of the final size distribution, the results of which are now shown in Figure 3.** We can confirm that two ‘shoulders’ are visible to the left of the main peak, both in PBS (Figure 3c) and in serum (Figure 3d). The first shoulder (~7 nm) corresponds to a certain amount of monomer that is still present in the aggregated samples. The nature of the second peak (~18 nm) is not exactly known, but must stem from smaller aggregates containing about 15-20 proteins.

With regard to the difference in size distributions in Fig 4 between 7 h and 20 h, what we wanted to say is that there are no major changes in the extent of barrier permeability between both time points. The shape of the distributions is somewhat different, but not significantly considering the indicated error range (dashed lines). In addition, it should be noted that subtle differences between both distributions may arise from the fact that the

results at each time point are determined from a different set of animals. **We have clarified this point in the text on page 8.**

Reviewer #3

1. **“There is an extensive body of supplemental work along with the manuscript. Both are very well readable, and in the initial part of the paper it is easily combined reading through both, later in the presentation of results in Figures 3 and 4, the reader has to go forth and back between the two, and also between different supplemental figures, a lot, which considerably hinders reading. Can't supplemental figure 8a+b be included in figure 3? And sfig 9 and 11 in fig.4, or somehow else to improve reading (9c,d are only discussed after sfig 11)?”**

Thank you for your valuable comments. As our work is quite extended it is a challenge to be concise while being sufficiently clear. Following your suggestion, we now combined Figure 3 with Supplementary Figure 8, and Figure 4 is combined with Supplementary Figure 9a, 9b and 11. We hope this will improve the readability/clarity of the manuscript.

2. **“The acronym rFRAP (rectangular FRAP) is used without specification, both in manuscript (p17), as in supplement (note 1)”**

Thank you for pointing that out. rFRAP has been specified the first time that it is used on page 14 in the new version.

3. **“Supp. note 1: 'in order to speed up the fitting...': shouldn't this also improve the fitting by improving the signal to noise with a factor $\sqrt{\text{\#pixels in rectangle}}$? Or are integration times such that single-pixels noise levels are not a limiting factor in the fits?”**

Not exactly. In the very first implementation of our algorithm, we performed fitting of the tempo-spatial recovery equation to the 2-D pixel data in the time-lapse image series directly, see Suppl. Note 1. Despite the fact that the noise between pixels is relatively high (i.e. like in typical confocal images), this works perfectly fine since the fitting is performed to a very high number ($120 \times 120 \times 30 = 432000$) of data points. As explained in Suppl. Note 1 this is, however, a very time-consuming approach. Therefore, we decided to sacrifice spatial information by averaging over ring-shaped areas, as depicted in Suppl. Fig. 1b. By averaging pixels in these areas, the noise is very much reduced indeed, as can be seen from the very smooth fluorescence profiles in Suppl. Fig. 1c. But this comes at the expense of a substantial decrease in data points to which the fitting is performed (e.g. in case of 15 rings: $15 \times 30 = 450$). As it is hard to predict theoretically what is the optimal situation (i.e. noise reduction vs. reduction in number of data points), we tackled this question pragmatically by performing simulations with varying number of ring areas. From the results in Suppl. Fig. 2

we conclude that averaging the pixel intensities in 10 ring areas offers the same precision as in the original method with fitting to all pixel values.

As the raised issue essentially comes down to the question of what is the minimal signal to noise ratio, we have performed additional simulations and experiments that are shown in Suppl. Fig. 7. We find that a SNR of 2.4 is about the minimum to still be able to retrieve the expected size distribution.

- 4. “Supp note 4: The monodisperse single and double component distribution fittings already give an apparent polydispersity just due to the fitting uncertainties. So when is this polydispersity real and how can this be assessed? Is there a goodness of fit, or model test parameter that can be used on the experimental and MEM-fitted recovery results?”**

Likely the reviewer is referring to Supplementary Note 3 and Supplementary Figure 4. The issue of (apparent) polydispersity even in case of a perfectly monodisperse sample is inherent to the chosen approach to interpret the FRAP data in terms of (semi)continuous distributions of diffusion coefficients without a priori assumptions. If the data of the double component system would be fitted by a two-component version of our FRAP model, of course both components would be virtually exactly retrieved. For the purpose of this work, however, we want to analyse continuous systems with potential high polydispersity. For our applications we cannot and want not make any a priori assumption to interpret the data in terms of a number of discrete components, for the simple reason that it would not make any sense to do so for the envisaged applications. That is why we use a mathematical framework (MEM analysis) which retrieves the best fit solution ($\chi^2 = 1$) to the data under condition of maximum entropy (i.e. with as little features as possible). In other words, from all the possible best fit solutions with $\chi^2=1$, MEM chooses that solution that has maximum entropy. So, for a two component system, rather than retrieving two delta-functions, it returns the most continuous solution that fulfils the χ^2 requirement for the data at hand.

We kindly refer the reviewer to our answer to comment 4 of Reviewer 2 for further information on the resolution of our method.

Also, please note that these considerations are not unique to our technique, but apply to any type of measurement where distributions are determined without prior assumptions, e.g. DLS or SEC. Every system has its own response function, and so does our cFRAP method.

- 5. “caption SFig 1: 't=1.6, 10.6, and 50.6 s', latter should be 6.4 and 33.4 according to figure.”**

Thank you for noticing that. The time points were correctly indicated in the new version.

6. "Sfig4 e,f: can these be compared by plotting it together with results for $T=\tau_{\text{slow}}$ and $\Delta t=0.5\tau_{\text{fast}}$?"

Suppl. Fig. 4 was adapted according to the reviewer's suggestion.

7. "On the cFRAP to DLS comparison: are the concentration the same? DLS needs very dilute suspensions (0.5mg/ml) according to Methods. It is not clear what the cFRAP concentrations are, I only could find stated 80mg/ml for the solution before injection into the animals. FRAP requires sufficient labelled species in the bleaching area, so do they measure the same diffusion coefficients or do we already see a concentration dependent correction for the FRAP result. Different concentration ranges that can be addressed could also be discussed in the discussion."

The same concentration of 0.5 mg/ml was used for both cFRAP and DLS measurement. **We added one sentence in the caption of Supplementary Figure 8 to clarify this.**

With regard to a potential concentration dependent diffusion coefficient we refer to the extra experiments that we performed on request of Reviewer 1 (Suppl. Fig. 7). Here, cFRAP experiments were performed on solutions with increasing concentration of FD40 from 4 to 2500 ug/ml. As the results in Suppl. Fig. 7 show, there was no influence of the concentration on the resulting size distribution.

8. "Sfig 9c, d: please use same numbering for same fluids in c and d. In legend for c ileum is misspelled."

The same color and numbering of the same biological fluids were revised in the new version. The typo in the legend was corrected.

9. "Captions Sfig 10 and 11: shock is misspelled"

The typo was corrected in the new version.

10. "Page 10, discussion: on cFRAP vs DLS: incorpoting the full spatial information in cFRAP improves the FRAP result itself, or maybe better it removes the error made by not including it. I do not see why this improves it compared to DLS, this is more a matter of averaging between both techniques. And maybe probing slightly different diffusion coefficients, free self diffusion in DLS and with concentration-dependent correction in FRAP."

We kindly refer the reviewer to our answer on Comment 4 by Reviewer 2 on this matter.

REVIEWERS' COMMENTS:

Reviewer #1 (Remarks to the Author):

In this reviewer's opinion, the authors have very carefully and completely addressed the comments of all the reviewers, and the paper is now acceptable.

Reviewer #2 (Remarks to the Author):

The authors addressed the reviewers' comments appropriately, though they did not give any justification on the use of rectangular photobleached area instead of circular counterpart. As shown in this paper, the rectangular one works fine for the purpose. However, it is desirable to point out advantages to use it.

Reviewer #3 (Remarks to the Author):

Xiong et al have submitted an extensive revision of their manuscript. The authors have thoroughly dealt with the criticism brought up by the reviewers. Several of the issues raised were shared among reviewers or pointed in the same direction, but these have been convincingly explained and the additional data has strengthened the manuscript. I also think that readability of the manuscript improved. I have no further comments and feel that the manuscript in its present form should be published in Nature Communications.

Response to the referees comments

We thank the editor and reviewers for careful reading of our rebuttal letter and the revised manuscript. We were pleased to read that all three reviewers find the changes made satisfactory for publication in Nature Communications. It seems, however, that we missed out on one question by Reviewer 2. Please find our response below.

Comment Reviewer 2: The authors addressed the reviewers' comments appropriately, though they did not give any justification on the use of rectangular photobleached area instead of circular counterpart. As shown in this paper, the rectangular one works fine for the purpose. However, it is desirable to point out advantages to use it.

The reason why we chose a rectangular format is because, to the best of our knowledge, it is the only geometry for which a full analytical solution is available that describes the recovery process both in time and space without any constraints on the size of the bleach area or recovery time by taking into account all the necessary parameters like the effective bleach resolution, the imaging resolution, etc. We refer to our recent tutorial-review on FRAP for a detailed account on the comparison of various published FRAP models. Based on our analysis the rectangle bleach method offers currently the most complete description of the spatio-temporal recovery process. Furthermore, in our opinion it was important to go for a model for which an analytical equation is

available as it will make calculations faster (as compared to numerical simulations) and will make the method easily accessible to other researchers. **This information is now added to the discussion of the revised manuscript on page 11.**